# Peroxisome Proliferator Activator α Agonist Clofibrate Induces Pexophagy in Coconut Oil-Based High-Fat Diet-Fed Rats

**DOI:** 10.3390/biology13121027

**Published:** 2024-12-07

**Authors:** Kanami Ohshima, Emika Hara, Mio Takimoto, Yidan Bai, Mai Hirata, Wen Zeng, Suzuka Uomoto, Mai Todoroki, Mio Kobayashi, Takuma Kozono, Tetsuhito Kigata, Makoto Shibutani, Toshinori Yoshida

**Affiliations:** 1Laboratory of Veterinary Pathology, Cooperative Department of Veterinary Medicine, Tokyo University of Agriculture and Technology, 3-5-8 Saiwai-cho, Tokyo 183-8509, Japanmshibuta@cc.tuat.ac.jp (M.S.); 2Cooperative Division of Veterinary Sciences, Tokyo University of Agriculture and Technology, 3-5-8 Saiwai-cho, Tokyo 183-8509, Japan; 3Smart-Core-Facility Promotion Organization, 3-5-8 Saiwai-cho, Tokyo 183-8509, Japan; tkozono@go.tuat.ac.jp; 4Laboratory of Veterinary Anatomy, Cooperative Department of Veterinary Medicine, Tokyo University of Agriculture and Technology, 3-5-8 Saiwai-cho, Tokyo 183-8509, Japan; fq6451@go.tuat.ac.jp

**Keywords:** clofibrate, high-fat diet, pexophagy, steatosis

## Abstract

Toxicologists know that peroxisome proliferator-activated receptors α agonists can induce peroxisomal proliferation in the liver of rodents but do not know whether pexophagy follows peroxisome proliferation. We showed that clofibrate, a peroxisome proliferator-activated receptors α agonist, induced pexophagy in the high-fat diet-fed rats through the pexophagy regulators. These novel findings suggested that rodent peroxisome proliferation with peroxisome proliferator-activated receptors α agonists is useful for understanding the biology of pexophagy in mammals and impacts the research on selected autophagy.

## 1. Introduction

In recent years, the incidence of nonalcoholic fatty liver disease (NAFLD) unrelated to alcohol consumption has increased worldwide [1,2]. Ten to twenty percent of NAFLD cases are simple fatty livers without inflammation or fibrosis. Meanwhile, 80–90% of cases are classified as nonalcoholic steatohepatitis (NASH) with inflammation and fibrosis [3,4]. The development of NAFLD is related to irregular lifestyles, such as the consumption of a high-sugar and -fat diet and chronic physical inactivity, and genetic factors, such as dysregulated lipid metabolism, diabetes mellitus, and insulin resistance [5]. Furthermore, hepatocellular injury due to DNA damage and increased oxidative stress increases the deposition of lipid droplets and persistent chronic inflammation, leading to NASH, which progresses to cirrhosis and increases the risk of liver cancer. However, their treatment is limited to exercise therapy, and appropriate drug therapy has not yet been developed [6].

Autophagy is an intracellular maintenance system that performs quality control of damaged or aged intracellular organelles and recycles intracellular components through selective self-digestion [7,8]. Autophagy is essential for securing nutrition for starving cells, maintaining cellular homeostasis, and inhibiting carcinogenesis. Cargoes, such as damaged or aged intracellular organelles, are explicitly selected by receptor proteins, enabling selective macroautophagy [9]. Pexophagy is a type of autophagy that selectively removes excessive or damaged peroxisomes [10]. Peroxisomes are essential in the β-oxidation of fatty acids, primarily degrading very long-chain fatty acids (>20 or 22 carbons), which leads to further β-oxidation, degrading short-chain fatty acids (<8 carbons) by the mitochondria [11]. Peroxisome biogenesis is highly complicated and is regulated by more than 30 peroxisomal membrane proteins, such as peroxisomal biogenesis factor (Pex), some of which regulate pexophagy [11,12]. For instance, Pex5 knockout mice showed an accumulation of abnormal mitochondria and excess fat droplets through pexophagy dysfunction and lipid metabolism, resulting in cerebral and liver damage [13]. As an initial step in pexophagy flux, Pex5 is mono-ubiquitinated by the E3 ubiquitin ligase Pex2. It is recognized by the cargo receptor p62/SQSTM1 or the neighbor of BRCA1 gene 1 (NBR1) and binds to microtubule-associated protein 1A/1B-light chain 3 (LC3) on sequestered membranes to form autophagosomes. Pex14 has been reported to bind to LC3 and tether peroxisomes to autophagosomes [9]. Autophagy-related proteins, including Atg3, Atg5, and Atg7, also regulate autophagosome formation [14]. Autophagosomes formed by these processes might combine with lysosomes as a late step in the autophagic flux to become autolysosomes, with the internal defective peroxisomes being degraded, followed by the reuse of degradation products in cellular metabolic pathways. Although the interaction between mitochondrial dynamics and mitophagy plays a critical role in NAFLD development [15,16], the interworking of peroxisome proliferation with pexophagy has not been thoroughly examined.

Peroxisome proliferator-activated receptors (PPARs) are metabolic nuclear receptors associated with the homeostatic regulation of blood glucose, triglycerides, and cholesterol [17,18]. There are three subtypes of PPARs in mammals: PPARα, PPARβ/δ, and PPARγ [19,20]. PPARα is distributed in tissues with a high capacity for fatty acid catabolism, such as the liver and kidney, and is responsible for the regulation of gene expression related to fatty acid metabolism, i.e., β-oxidation and triglyceride transport [19,20]. PPARγ in white and brown adipose tissues plays a central role in inducing adipocyte differentiation and enhancing fat storage in mature adipocytes. In contrast, PPARβ/δ is universally expressed in various organs and tissues, including the skeletal muscle. In the liver, PPARα predominates under fasting conditions, leading to progressive β-oxidation and ketone body synthesis, whereas PPARγ predominates under feeding conditions, enhancing lipid synthesis [21]. Clofibrate (CF) is a PPARα agonist that induces peroxisomal proliferation and promotes the β-oxidation of fatty acids in the liver of rodents [22]. In mice, the number and average size of peroxisomes doubled and lipid droplets decreased by one-third after four days of 0.5% CF administration in the diet [23]. In rats, clofibric acid (130 mg/kg), the active metabolite of CF, increased the mass of peroxisomal membranes in the liver [24]. Fenofibrate, a PPARα agonist, also significantly increased the expression of peroxisomal genes and proteins involved in peroxisomal biosynthesis and function [25]. The inhibition of peroxisomal fatty acid oxidation by a high-fat diet (HFD) was also significantly restored by the coadministration of fenofibrate. Because fenofibrate has potential therapeutic effects in NAFLD due to its ability to reduce liver lipid accumulation and its antioxidant, anti-inflammatory, and antifibrotic properties, CF may also be effective in treating NAFLD [26].

Fatty liver-related early carcinogenesis has been extensively studied using a two-step hepatic carcinogenesis model in rats fed with an HFD [27,28,29,30,31,32]. Coconut oil, principally comprising medium-chain fatty acids (8–12 carbons), is readily absorbed and metabolized to convert to ketone bodies, the final metabolites of fatty acids, which are subsequently excreted. Unlike long-chain fatty acids, medium-chain fatty acids are less likely to accumulate as lipid droplets, making coconut oil a diet choice for patients with obesity, dyslipidemia, and hypertension [33,34]. In this study, we aimed to evaluate the potential of a coconut oil-based HFD on steatosis, hyperlipidemia, and the development of precancerous lesions in a rat model of hepatic carcinogenesis and determined whether the PPARα agonist CF promotes peroxisome proliferation in hepatocytes to mitigate steatosis via enhanced β-oxidation. We hypothesized that the excess peroxisomes generated during this process are subsequently degraded by pexophagy. To this end, we assessed the effects of CF on pexophagy in this rat model using immunohistochemistry for key pexophagy and steatosis markers as follows: the pexophagy receptor NBR1, selective autophagy receptor p62, pexophagy indicator Pex5, lysosomal indicator lysosome-associated membrane protein 2 (LAMP2), and adipose differentiation-related protein (ADRP). Additionally, transmission electron microscopy (TEM) was applied to observe autophagosomes and autolysosomes, providing ultrastructural insights into pexophagy flux.

## 2. Materials and Methods

### 2.1. Chemicals

N-diethylnitrosamine (DEN; CAS No. 55-18-5, purity > 99%) and CF (CAS No. 637-07-0, purity > 98%) were purchased from Tokyo Chemical Industry Co., Ltd. (Tokyo, Japan) and FUJIFILM Wako Pure Chemical Co., (Osaka, Japan), respectively.

### 2.2. Animal Experiment

A total of 25 5-week-old male F344/DuCrlCrlj rats were purchased from Charles River Laboratories Japan, Inc., (Kanagawa, Japan) and reared using a powdered diet (MF, Oriental Yeast Co., Tokyo, Japan) and an accessible water supply at a temperature of 23 ± 3 °C and a humidity of 50 ± 20% with a 12 h light/12 h dark lighting cycle in a clean rack with paper-type rat enrichment, with a maximum of three rats per cage. All rats were maintained on a basal diet during the one-week acclimation period. According to established methods [27], a two-step (initiation and promotion) hepatic carcinogenesis model was established (see Appendix A for details; Appendix A). At the beginning of the study, all rats underwent initiation treatment by the intraperitoneal administration of DEN (200 mg/kg body weight) and were started on a basal or HFD (D12331, 58 kcal% fat including coconut oil; Research Diets, Lane, NJ, USA). From the second week of DEN administration, the rats were divided into four groups of six or seven rats: a basal diet-fed group (CTL group, six rats), basal diet-fed and CF-administered group (CF group, six rats), HFD containing coconut oil-fed group (HFD group, six rats), and combined HFD and CF administration group (HFD + CF group, seven rats). CF was administered at a concentration of 0.3%, according to a previous report [35], for 12 weeks until the necropsy. Three weeks after the DEN administration, all animals underwent 2/3 partial hepatectomy to increase cell proliferation under deep isoflurane anesthesia by an experienced surgeon (T.Y.); one in the CTL group died on the same day because of partial hepatectomy. During the study period, the general condition of the animals was observed daily, and body weight and food and water intake were measured weekly. At 14 weeks after DEN administration, after overnight fasting (water supply continued) [23], all animals were abdominally opened under isoflurane anesthesia, blood was collected from the abdominal aorta using heparinized syringes, and they were euthanized by blood release from the abdominal aorta and vena cava. The livers were collected and fixed in 4% paraformaldehyde phosphate buffer (PFA; pH 7.4) for histopathological examination and immunohistochemical staining. A portion of the liver was frozen in liquid nitrogen and stored at −80 °C. The animal experiment protocol was reviewed and approved by the Experimental Animal Committee of the Tokyo University of Agriculture and Technology (Approval No.: R03-150). The study was conducted in compliance with the ARRIVE guidelines.

### 2.3. Blood Biochemistry

Plasma was separated from blood collected at the end of the study and analyzed for aspartate aminotransferase (AST), alanine aminotransferase (ALT), total cholesterol (T-CHO), free cholesterol (F-CHO), esterified cholesterol (E-CHO), esterified cholesterol-total cholesterol (E/T) ratio, triglyceride (TG), non-esterified fatty acid (NEFA), low-density lipoprotein cholesterol (LDL-C), high-density lipoprotein cholesterol (HDL-C), blood urea nitrogen (BUN), and creatinine (CRE) with a 7180 Clinical Analyzer (Hitachi, Ltd., Chiyoda-ku, Tokyo, Japan) in Oriental Yeast Co., Ltd. (Shiga, Japan).

### 2.4. Histopathology

Hematoxylin-eosin staining was performed on PFA-fixed liver after thin sectioning to approximately 3 μm (see Appendix A for details). In the liver, based on the NAFLD activity score [29,36,37] and international terminology for the rodent liver [38], hepatocytes were scored for steatosis, balloon-like changes, inflammatory cell foci, and acidophilia, and a total score was calculated.

### 2.5. Ultrastructural Examination

Fresh liver pieces were fixed in a 2.5% glutaraldehyde solution, followed by post-fixation with 1% osmamic acid (Nisshin EM, #300, Tokyo, Japan) (see Appendix A for details). The samples were embedded in epoxy resin (TAAB, #T024, Aldermaston, UK), ultra-sectioned to 80 nm, stained with EM stainer (Nisshin EM, #311, Tokyo, Japan) and lead, and observed with a transmission electron microscope (TEM), JEM-1400Flash (JEOL Ltd., Tokyo, Japan). TEM analysis was performed at Tokyo University of Agriculture and Technology for Smart-Core-Facility Promotion Organization.

### 2.6. Immunohistochemistry

PFA-fixed livers were thinly sliced and subjected to immunohistochemistry (IHC). The antibodies used, antigen activation method, and antibody dilution conditions are listed in Appendix A. Signal detection was performed according to the protocol of the VECTASTAIN^®^ Elite ABC Kit (Vector Laboratories Inc., Burlingame, CA, USA). After visualization, the tissues were counterstained with hematoxylin (see Appendix A for details). Immunostaining was performed for preneoplastic liver lesions positive for glutathione S-transferase placental form (GST-P) and surrounding hepatocytes for Pex5, NBR1, ADRP, p62, and LAMP2. For GST-P-positive precancerous lesions, the number and area of lesions >0.1 mm in diameter and total liver area per were calculated using Fiji (2.16.0, https://imagej.net/software/fiji/downloads (accessed on 15 May 2022)) [27]. Granules that expressed Pex5, NBR1, ADRP, LAMP2, and p62 in hepatocytes were measured for positivity rate per unit area using Fiji.

### 2.7. Real-Time Reverse Transcrip Tion-Polymerase Chain Reaction Analysis

mRNA was extracted from liver samples from five rats in the CTL group, six each in the CF and HFD groups, and seven in the HFD + CF group. Real-time reverse transcription–polymerase chain reaction (RT-PCR) was conducted using selected primers (Appendix A), as previously reported (see Appendix A for details) [30,39]. The mRNA expression levels of each gene were measured using the hypoxanthine phosphoribosyl transferase 1 gene as an endogenous control.

### 2.8. Statistical Analyses

Means and standard deviations were calculated for all data. Statistical analysis was performed using Tukey’s or Steel–Dwass test, and a significance level of 5% or less was considered significant. The cumulative distribution of the preneoplastic lesion area between the two groups was statistically analyzed using the Kolmogorov–Smirnov test. The obtained *p*-values were adjusted using the Bonferroni correction to compare the four groups.

## 3. Results

### 3.1. Feeding a Coconut Oil-Based HFD Increases Intra-Abdominal Adipose Tissue Weight, and the HFD Combined with CF Administration Further Increases Liver Weight

To elucidate the effects of obesity and organ weight with CF administration, HFD feeding, and their combination, we measured body weight changes during the study and the weights of the liver and intra-abdominal fat at autopsy. The coconut oil-based HFD group showed a significantly reduced weight gain during weeks 1–4 but almost identical levels as the CTL group after week 5 (Appendix A). The final body weights of the HFD group at autopsy were similar to those of the CTL group (Table 1). However, food intake was significantly decreased in the HFD groups (HFD and HFD + CF groups) compared with the CTL group (Table 1 and Appendix A), probably due to food preference. Thus, the HFD did not induce obesity, as shown by the body weight; however, it significantly increased the absolute and relative intra-abdominal fat weights (Figure 1A,B). The CF groups (CF and HFD + CF groups) showed a significant inhibition of body weight gain during the study (Appendix A). Their final body and absolute intra-abdominal fat weights were significantly lower than those of the CTL and HFD groups (Table 1; Figure 1A,B). Water intake was not significantly different among all groups (Table 1; Appendix A). At autopsy, the absolute and relative liver weights of the liver were significantly higher in the CF group than in the CTL group by 1.4- and 1.6-fold, respectively (Figure 1C,D), as previously reported [40], with an even more significant increase in the HFD + CF group by 1.8- and 2.0-fold, respectively. The increased liver weights in the HFD + CF group were consistent with previously reported findings in rats fed a cafeteria diet (HFD + high-sucrose diet) followed by two-week CF administration (0.1 and 0.25%) [41]. These results indicate that the HFD increased intra-abdominal fat without affecting body weight. In contrast, CF inhibited body and intra-abdominal fat weight gain but increased liver weight, especially in HFD-fed rats.

### 3.2. Coconut Oil-Based HFD Increases Marginal Hepatotoxicity; However, HFD Combined with CF Administration Inhibited the Effects

To elucidate hepatotoxicity and lipidemia with CF administration, HFD feeding, and their combination, we conducted blood biochemistry in each group. Concerning markers of liver injury, AST showed an increasing trend in the CF and HFD groups compared with the CTL group by 1.5- and 1.8-fold, respectively, whereas ALT showed an increasing trend only in the HFD group by 2.2-fold (Table 2); these effects were not evident in the HFD + CF group. These findings suggest that the marginal hepatotoxicity induced by HFD was suppressed when HFD was combined with CF administration, as previously reported for fenofibrate [42,43]. TG showed a decreasing trend in the CF and HFD + CF groups compared with the CTL group (Table 2).

### 3.3. Coconut Oil-Based HFD Increases NAFLD Scores; However, HFD Combined with CF Administration Inhibits the Effects

To elucidate the effects of CF administration, HFD feeding, and their combination on steatosis, we examined the liver with histopathology and immunohistochemistry for the fat droplet membrane. The histopathological examination of the liver showed significantly increased NAFLD activity, steatosis, and ballooning change scores (excluding the inflammation score) in the HFD group compared with the CTL group (Figure 1E–I). Fatty degeneration of the hepatocytes (steatosis) was not observed in the CTL and CF groups, but was mildly observed in the HFD group. IHC staining also showed that HFD increased the percentage of cells positive for ADRP (Figure 1J,K). These effects on hepatocyte changes with HFD were inhibited by the coadministration of CF, with a significantly increased score of hepatocyte acidophilia (Figure 1E,L), related to peroxisome proliferation [40,41,42,43].

### 3.4. Coconut Oil-Based HFD Increases NBR1-Positive Granules, and HFD Combined with CF Administration Increased Pex5-, p62-, and LAMP2-Positive Granules

To elucidate pexophagy with CF administration, HFD feeding, and their combination, we conducted immunohistochemistry for autophagy markers. The HFD group had a 2.4-fold increase in the number of NBR1-positive granules compared with the CTL group (Figure 2A,B). The CF group had increased NBR1- and p62-positive granules (1.3- and 4.2-fold increases, respectively) (Figure 2A–D) and significantly increased LAMP2 (4.9-fold increase) compared with the CTL group (Figure 2E,F). The HFD + CF group had significantly decreased NBR1-positive granules and increased p62-, LAMP-2-, and Pex5-positive granules (7.5-, 7.2-, and 71.4-fold increases, respectively) compared with the HFD group (Figure 2A–H). These results suggest that the involvement of pexophagy was more clearly observed in the HFD + CF group than in the HFD group.

### 3.5. Coconut Oil-Based HFD and CF Administration Induce Phagophores

To elucidate peroxisome proliferation and pexophagy with CF administration, HFD feeding, and their combination, we conducted TEM using liver samples in each group. The ultrastructural examination revealed that the HFD group had an increased number of fat droplets and the enlargement of mitochondria in hepatocytes (Appendix A). Peroxisomes with dense cores and mitochondria were evident in the CF group (Figure 3A). Organelles, possibly peroxisomes, were sometimes surrounded by sequestered membranes, indicating phagophore expansion (Figure 3B and Appendix A). In the HFD + CF group, many peroxisomes and mitochondria, with organelles surrounded by isolated membranes and autolysosomes containing organelles, were observed (Figure 3C,D and Appendix A).

### 3.6. Coconut Oil-Based HFD and CF Administration Affect the Development of Preneoplastic Liver Lesions

Previous studies demonstrated that CF administration inhibited GST-P-positive foci [40], while HFD feeding increased them in the early hepatocarcinogenesis model of rats [28,29,30,31,32]. We elucidated the effects of CF administration, HFD feeding, and their combination on the number and area of GST-P-positive preneoplastic lesions. The HFD group had an increased number but not the area of GST-P-positive foci compared with the CTL group, although the difference was not significant (Figure 4A–C). CF administration did not affect the number and area of GST-P-positive foci; however, the number was significantly decreased and the area was significantly increased in the HFD + CF group compared with those in the HFD group. Examination of the size distribution of GST-P-positive foci in each group revealed that the HFD group had relatively small GST-P-positive foci (Figure 4D). In contrast, the HFD + CF group tended to have larger GST-P-positive foci, indicating that the size of GST-P-positive foci varied (Figure 4D). Based on these results, we concluded that combined HFD and CF administration marginally increased hepatic precancerous lesions under these experimental conditions.

### 3.7. Coconut Oil-Based HFD Combined with CF Administration Altered the Expression of Autophagy-, Lipid Metabolism-, and Oxidative Stress-Related Genes

We examined the expression of autophagy, lipid metabolism, and oxidative stress genes in the liver to compare histopathological and immunohistochemical changes. Regarding the expression levels of autophagy-related genes, significant increases were observed in *P62* and *Pex14* in the CF group compared with those in the CTL group (Figure 5). Compared with the HFD group, the HFD + CF group showed a significant increase in the autophagy regulatory factor gene *Atg7*, the autophagosome membrane protein gene *Lc3*, the pexophagy-specific receptor protein genes *Nbr1* and *P62*, and the peroxisome membrane protein genes *Pex2* and *Pex14*.

Regarding the expression levels of genes related to lipid metabolism, compared with the CTL group, the CF group showed significant increases in the cholesterol and phospholipid transporter gene *Abca1*, the genes *Ppara* and *Pparg* for transcription factors that promote β-oxidation, the fatty acid synthase gene *Scd1*, and the gene *Srebf1* for sterol transcription factors (Figure 6). The HFD group also showed significant increases in the expression of the fatty acid desaturase gene *Acox2* and the fatty acid synthase gene *Fasn*. In contrast, the HFD + CF group showed substantial increases in *Abca1*, *Acox1*, *Acox2*, *Fasn*, *Ppara*, *Pparg*, *Scd1*, and *Srebf1* expression compared with the CTL group. The expression levels of *Abca1*, *Acox1*, *Fasn*, and *Srebf1* were also significantly higher in the HFD + CF group than in the HFD group.

The expression levels of the oxidative stress-related genes *Catalase* and *Sod2* were significantly higher in the CF and HFD + CF groups than in the CTL group (Figure 7). There was a significant decrease in the expression of the antioxidant enzyme gene *Sod3* in the CF group compared with that in the CTL group.

These results showed that CF administration affected several genes involved in autophagy, lipid metabolism, and oxidative stress in normal diet-fed rats and further affected those in HFD-fed rats.

To elucidate the relationship between lipidemia and hepatic gene expression, we also determined the correlation between selected plasma cholesterol levels and hepatic gene expression levels. A high correlation was detected between T-CHO, F-CHO, E-CHO, E/T, LDL-C, and HGL-C and lipid metabolism and antioxidant response genes (Appendix A).

## 4. Discussion

The study aimed to determine the effects of a coconut oil-based HFD in an early liver carcinogenesis rat model and clarify steatosis and pexophagy when combined with PPARα CF administration in this model. We previously reported that a fatty liver-related early liver carcinogenesis model was generated by an HFD containing lard and confirmed the development of steatosis and preneoplastic lesions in the liver [28,29,30,31,32]. We selected male F344 rats in these studies, as male rats were sensitive to female rats in liver carcinogenicity studies [27,44]. Male patients with NAFLD have a higher risk of steatosis, NASH, fibrosis, and HCC than females, as estrogen can protect against NAFLD [45]. In this study, we confirmed that mild obesity (increased intra-abdominal fat), liver injury, and steatosis were detected in rats fed a coconut oil-based HFD; however, the effects on hyperlipidemia were not evident. In addition, we demonstrated that HFD combined with CF causes pexophagy following peroxisome proliferation using IHC and TEM. The results suggest that steatosis was completely suppressed through peroxisome activation; however, an imbalance between the induction and inhibition of pexophagy was demonstrated by increased levels of p62, LAMP2, and Pex5.

Coconut oil, containing mainly lauric acid, a medium-chain fatty acid, has been widely used to prevent obesity and hyperlipidemia in studies of metabolic syndrome in a rodent model. For instance, the forced oral administration of medium-chain fatty acids to mice fed a 45% HFD for 12 weeks prevented increases in plasma TG, T-CHO, and LDL-C levels and improved steatosis by inhibiting the induction of fatty liver-related proteins SREBP2, SREBP1c, FAS, and ACC in the liver of obese mice [46]. Similar experimental results have been demonstrated in a rat study: coconut oil supplementation for 10 weeks prevented increases in lipid levels and AST in the serum, steatosis score, and TG content in the liver [47]. Compared with the CTL group, the HFD group showed a decreasing trend in TG but no apparent effect on F-CHO, NEFA, or HDL-C and an increasing trend in AST and ALT, without adverse effects on liver weight or histopathological examination of the liver. The results suggest that the effects of HFD on serum lipids and liver function in blood tests were mild; however, steatosis was detected by hematoxylin-eosin staining and immunostaining for ADRP. This was supported by the findings of the gene expression analysis of the liver, where HFD increased the expression of *Fasn*. This enzyme synthesizes palmitic acid from malonyl CoA and facilitates fatty acid synthesis, as does *Acox2*, an acetyl CoA oxidase involved in the β-oxidation of molecular chain fatty acids [48,49,50]. Increased *Fasn* expression might be associated with coconut oil-induced pathogenesis (increased fat droplets). In contrast, an increase in *Acox2* might be an inhibitory response to fatty liver and may reflect disease suppression by coconut oil. Notably, no significant changes in the antioxidant genes in the HFD group suggest that steatosis-mediated oxidative stress was obscured in rats fed a coconut oil-based HFD.

The effects of PPARα agonists have been widely examined in rodent livers. CF specifically increases peroxisomes, induces hepatocyte hypertrophy and acidophilia, and increases peroxisome-related enzymes in the livers of rats fed normal diets [51,52]. Similar results were confirmed in this study, as shown by the inhibitory effect of CF on steatosis and the increased liver weight and hepatocyte acidophilia, along with peroxisome proliferation in TEM analysis. The effects were accompanied by increases in the gene expression of *Abca1*, *Acox1*, *Acox2*, *Fasn*, *Ppara*, *Pparg*, *Scd1*, *Srebf1*, *Catalase*, and *Sod2* when compared with the control. Sterol regulatory element-binding protein 1 (Srebf1) is a transcription factor related to fatty acid metabolism that promotes the transcription of Abca1, involved in HDL formation. Fasn, PPARγ, and Scd1 play a role in the fatty acid synthesis and unsaturation, and Pparα promotes the transcription of *Acox1* and *Acox2* in β-oxidation [48,49,50]. Furthermore, Sod2 and catalase play roles in synthesizing hydrogen peroxide from superoxide and hydrogen peroxide degradation, respectively [49]. The results suggest that the interaction between fatty acid synthesis and degradation was detected by gene expression analysis in the liver when CF inhibited steatosis, even in basal diet-fed rats.

We demonstrated that CF inhibits steatosis in HFD-fed rats. Under HFD, numerous fat droplets were formed in hepatocytes, as shown by ADRP immunostaining and TEM analysis. It can be inferred that CF administration under these conditions promoted more CF-induced peroxisome proliferation than in the CTL group. Although we did not measure lipid content in the liver, Sugatani et al. demonstrated that liver triacylglycerol levels were significantly reduced in a cafeteria diet (HFD + high-sucrose diet) followed by a two-week CF administration (0.1 and 0.25%) [41]. A similar study was conducted on rats fed a high-fructose diet with fenofibrate [42]. Gene expression related to fatty acid synthesis and β-oxidation was enhanced, particularly that of *Abca1*, *Acox1*, *Fasn*, *Scd1*, and *Srebf1*. Similar data have been reported when CF or CF analogs were administered to rats fed an HFD [41,53], suggesting that the effects of CF differ depending on the presence or absence of fatty liver. The genes for *Catalase* and *Sod2*, proteins that exhibit antioxidant activity, were significantly increased in the CF and HFD + CF groups compared with the control and HFD groups. However, their expression levels were similar, suggesting that the expression of antioxidant enzymes, in relation to the increase in peroxisomes and oxidative stress, occurs under both dietary conditions. Notably, HFD causes an increase in malondialdehyde, an oxidative stress indicator, and CF administration suppresses this increase [54]. The antioxidant activity of CF has also been reported in an in vitro model of NAFLD following fatty acid exposure in rat hepatocellular carcinoma cells [55]. PPARα activation can protect against oxidative stress by expressing antioxidant and anti-inflammatory genes [26]. However, the effects of fibrates on oxidative stress markers, such as malondialdehyde, were not consistent; fenofibrate failed to reduce the liver content of rats fed a high-fructose diet [42]. As we demonstrated a strong correlation between plasma cholesterol levels and hepatic gene expression for lipid metabolism or antioxidant response, further studies are required on the regulation of gene expression in the liver and its effect on systemic lipid metabolism.

To our knowledge, studies on pexophagy in rodent livers are limited. In a previous study, CF was administered to rats for three days and then withdrawn for one or two days; many peroxisomes incorporated into autophagosomes were observed [56]. In this study, overnight fasting was performed from the evening of the day before the final autopsy to induce mild autophagy in the liver, as previously reported [28,29,30,31,32]. Under the study conditions, we quantitatively examined the expression levels of p62, Pex5, and LAMP2 in the liver. To explore autophagic flux clearly, we briefly summarize how to understand autophagy induction and inhibition. Because LC3 is expressed both inside and outside the sequestered membrane, it is impossible to determine whether autophagy is inhibited or promoted when its expression is enhanced. Therefore, to accurately determine autophagic flux, we must examine the levels of autophagy receptors and autophagy-related membrane proteins in targeted organelles that fuse to lysosomes as autophagy proceeds. The levels of receptors and organ proteins decrease during digestion in autolysosomes. In contrast, increased expression of these molecules suggests that autophagy is inhibited [7,8]. For example, increased expression of LC3-positive granules indicates induction or late inhibition of autophagy, whereas decreased expression suggests early inhibition of autophagy. In contrast, decreased expression of the autophagy receptor p62 indicates the induction of autophagy, whereas increased expression suggests early and late inhibition of autophagy. Furthermore, increased expression of LAMP1-positive granules suggests autophagy induction [57]. The combination of autophagy markers can be analyzed to predict autophagy flux on pathological specimens.

Our data suggest that pexophagy was induced in the CF group, as we demonstrated a mild increase in p62 and a clear increase in LAMP2 expression, with no difference in Pex5 and NBR1 expression in the CF-treated group compared with those in the CTL group. With autophagy induction, gene expression analysis showed that the levels of *p62* and *Pex14* were significantly increased in the CF group compared with those in the CTL group. Pex14 can directly bind to LC3-II, leading to pexophagy under starvation [11,58]. Therefore, our data suggest a pathway through which peroxisomes bind to autophagosomes and are incorporated into autophagosomes in the liver when CF is administered to rats. This was supported by the findings in TEM, in which organelles, possibly peroxisomes with phagophore expansion, and autolysosomes were often observed. Notably, the condition of pexophagy differed between the CF and HFD + CF groups. Increased NBR1 expression in the coconut oil-based HFD group suggests that pexophagy was suppressed. When comparing the HFD + CF group with the HFD group, there was a marked increase in Pex5 and p62 and a non-significant but clear increase in LAMP2. The marked increase in Pex5 and p62 levels may indicate that excess peroxisomes induced by CF administration remained unprocessed by pexophagy. However, the increase in LAMP2 suggests that some of the excessive peroxisomes were processed by pexophagy. Therefore, the current study condition in the HFD + CF group might exhibit excessive peroxisome proliferation and pexophagy processing. This interpretation may be supported by the marked increase in liver weight, along with numerous peroxisomes and pexophagy in the hepatocytes with TEM. Gene expression analysis of the liver showed that Pex2 was significantly increased in the HFD + CF group compared with that in the HFD group, a change that may be related to the enhanced mono-ubiquitination of Pex5 by Pex2 during autophagy [11,12]. We speculated that gene activation and executive protein expression on pexophagy might deal with excessive peroxisomes.

## 5. Conclusions

In the present study, we clarified the involvement of pexophagy in a steatosis model with rats fed coconut oil-based HFD by administering a PPARα agonist, CF. By immunohistochemically analyzing liver tissues with critical pexophagy markers, we demonstrated that CF might induce pexophagy in basal diet-fed rats (control). However, unprocessed peroxisomes remained due to the induction of pexophagy under the conditions of the HFD in combination with CF. The inhibitory effect of CF on fatty liver in both groups was evident, indicating that the conditions of the diets (basal diet or HFD) fed to the animals resulted in differential responses to pexophagy following peroxisome proliferation induced by CF administration. Our results suggest that when a PPARα agonist is treated in animals with hyperlipidemia and fatty liver, pexophagy occurs in the liver due to peroxisome proliferation, which may result in residual peroxisomal processing. Although we were unable to clarify the effect of these residual peroxisomes in this experiment, at least in the hepatic gene expression analysis, the expression of genes related to β-oxidation in the combined group was remarkable, and the expression of antioxidant genes to remove reactive oxygen species produced in the lipid metabolism process was similar to that in the CF group. These results suggest that the remaining peroxisomes maintained their physiological functions, but pexophagy might not be fully functional in steatosis models, resulting in a marginal increase in the preneoplastic lesions. Although peroxisomes and PPARα play a role in lipid metabolism, the role of pexophagy is not fully elucidated. The present study model is a powerful tool for understanding the biological processes involved in peroxisome proliferation and pexophagy in mammals.

## Figures and Tables

**Figure 1 biology-13-01027-f001:**
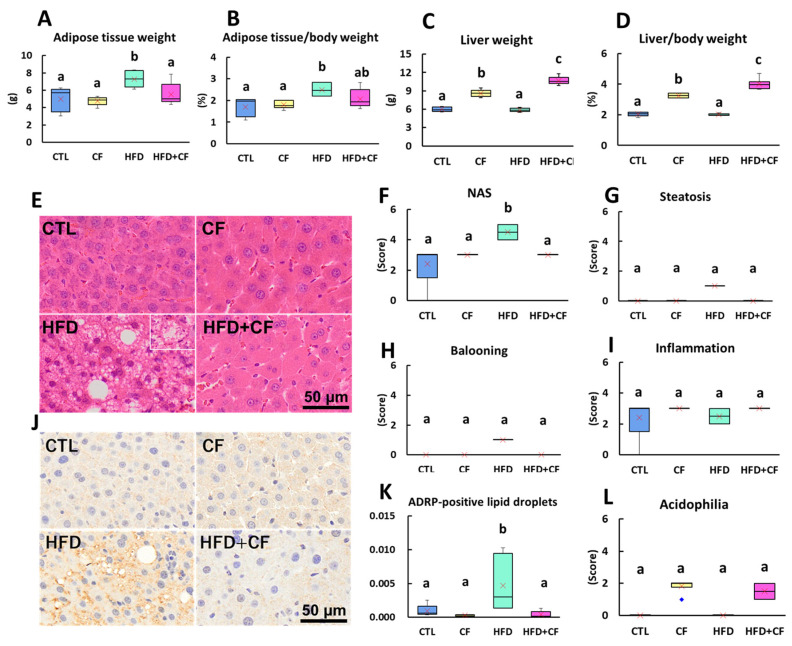
Organ weights and representative images and qualitative analyses of fatty liver of rats. (**A**) Absolute weight of intra-abdominal fat weight. (**B**) Relative intra-abdominal fat weight to body weight. (**C**) Absolute liver weight. (**D**) Relative liver weight to body weight. (**E**) Normal hepatocytes, slight fatty droplet deposition, and acidophilia and hypertrophy in hepatocytes in the CTL, CF, HFD, and HFD + CF groups, respectively. Bar = 50 μm. Hematoxylin and eosin. (**F**) NAFLD activity score (NAS). (**G**) Steatosis score. (**H**) Ballooning score. (**I**) Inflammation score. (**J**) ADRP-positive lipid droplets, evident in the HFD group. (**K**) Area of ADRP-positive lipid droplets. (**L**) Acidophilia score. Data represent a box-and-whisker plot (red cross, mean; blue dots, outliers). Group: CTL, rats received control diet (Blue); CF, rats received control diet mixed with clofibrate (Yellow); HFD, rats received high-fat diet (Light green); HFD + CF, rats received high-fat diet mixed with clofibrate (Light red). Different letters indicate significant differences between groups (*p* < 0.05, significantly different by Tukey’s or Steel–Dwass test).

**Figure 2 biology-13-01027-f002:**
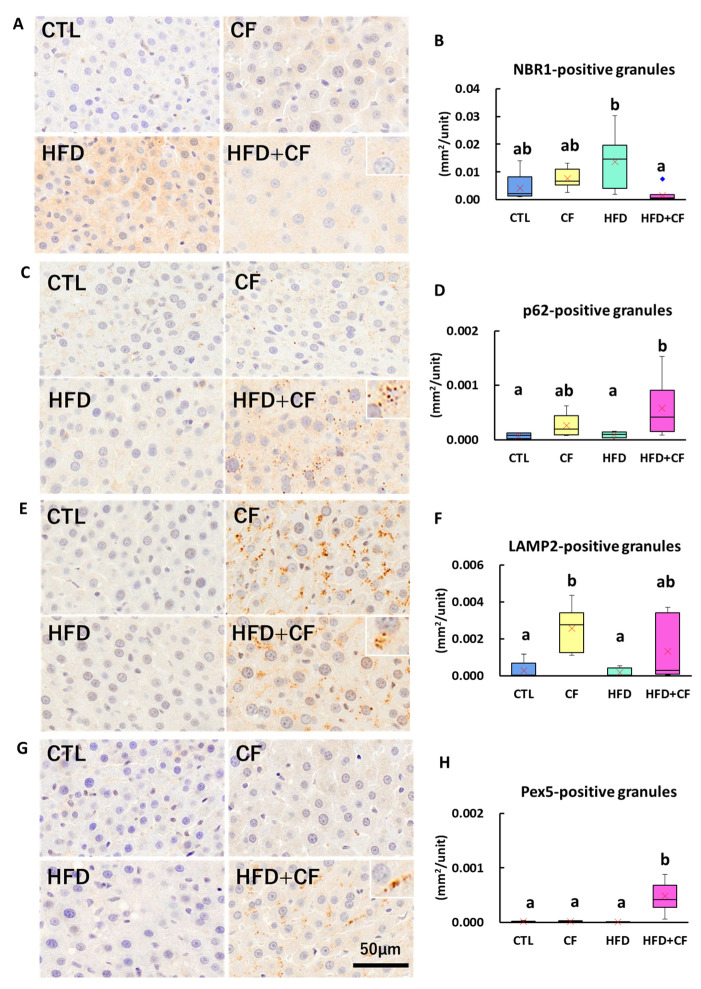
Representative images and qualitative analyses of autophagy-related markers in hepatocytes of rats. (**A**) Immunohistochemistry for NBR1-positive granules, counterstained with hematoxylin, in the CTL, CF, HFD, and HFD + CF groups. Bar = 50 μm. (**B**) Quantitative analysis of NBR1-positive granules. (**C**) Immunohistochemistry for p62-positive granules, counterstained with hematoxylin, in the CTL, CF, HFD, and HFD + CF groups. Bar = 50 μm. (**D**) Quantitative analysis of p62-positive granules. (**E**) Immunohistochemistry for LAMP2-positive granules, counterstained with hematoxylin, in the CTL, CF, HFD, and HFD + CF groups. Bar = 50 μm. (**F**) Quantitative analysis of LAMP2-positive granules. (**G**) Immunohistochemistry for Pex5-positive granules, counterstained with hematoxylin, in the CTL, CF, HFD, and HFD + CF groups. Bar = 50 μm. (**H**) Quantitative analysis of Pex5-positive granules. Insets: a higher magnification of positive granules in the HFD = CF group (**A**,**C**,**E**,**G**). Data represent a box-and-whisker plot (red cross, mean; blue dots, outliers). Group: CTL, rats received control diet (Blue); CF, rats received control diet mixed with clofibrate (Yellow); HFD, rats received high-fat diet (Light green); HFD + CF, rats received high-fat diet mixed with clofibrate (Light red). Different letters indicate significant differences between groups (*p* < 0.05, significantly different by Tukey’s or Steel–Dwass multiple comparison test).

**Figure 3 biology-13-01027-f003:**
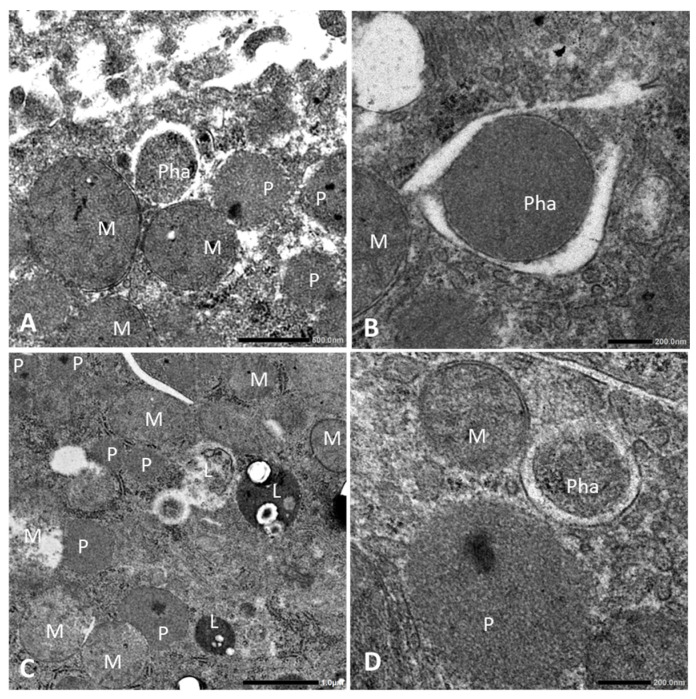
Representative TEM images of peroxisome and phagophores of hepatocytes. (**A**,**B**) Peroxisomes (P), mitochondria (M), and phagophore (Pha) in the CF group. An organelle, possibly peroxisome captured by phagophore (sequestered membranes), indicating phagophore expansion. (**C**,**D**) Peroxisomes, mitochondria, phagophore, and autolysosomes (L) in the HFD + CF group. Autolysosomes consist of organelles and lysosomes. TEM, transmission electron microscopy. Bar = 200 (**B**,**D**), 500 nm (**A**) and 1.0 μm (**C**).

**Figure 4 biology-13-01027-f004:**
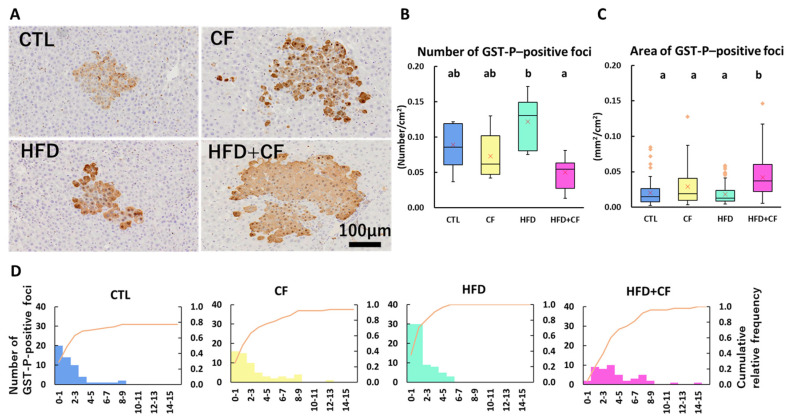
Representative images and qualitative analyses of GST-P-positive foci of rats. (**A**) GST-P-positive foci, counterstained with hematoxylin, in the CTL, CF, HFD, and HFD + CF groups. Bar = 100 μm. Box plot of number (**B**) and area (**C**) of GST-P-positive foci. (**D**) Distribution of GST-P-positive foci in each size and cumulative curve of GST-P-positive foci. Data represent a box-and-whisker plot (red cross, mean; yellow dots, outliers) (**B**,**C**). Group: CTL, rats received control diet (Blue); CF, rats received control diet mixed with clofibrate (Yellow); HFD, rats received high-fat diet (Light green); HFD + CF, rats received high-fat diet mixed with clofibrate (Light red). Different letters indicate significant differences between groups (*p* < 0.05, significantly different by Tukey’s or Steel–Dwass multiple comparison test). Kolmogorov–Smirnov test indicated that significant increases were detected between the CTL/CF/HFD and HFD + CF groups (*p* < 0.0083, modified with Bonferroni correction).

**Figure 5 biology-13-01027-f005:**
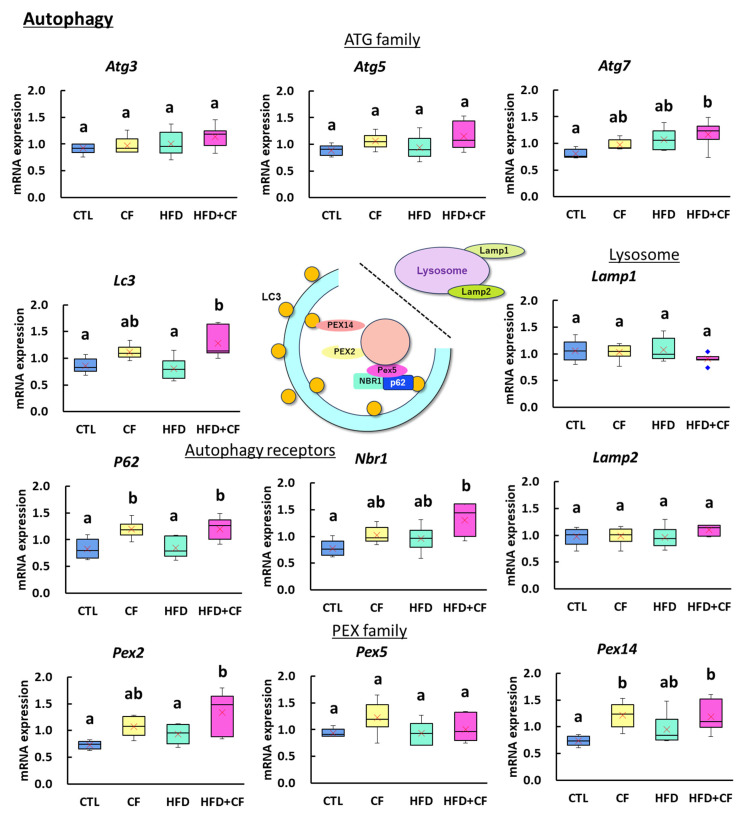
Gene expression analysis on autophagy in the liver. *Atg3*, *Atg5*, and *Atg7* are genes initiating phagophore formation, binding with the product of *Lc3*, a gene for autophagosome formation. *P62* and *Nbr1* are genes of receptors for cargos including damaged peroxisomes. *Lamp1* and *Lamp2* are genes of lysosome membrane proteins. *Pex2*, *Pex5*, and *Pex4* are genes for damaged peroxisomes, leading to pexophagy. Data represent a box-and-whisker plot (red cross, mean; blue dots, outliers). Group: CTL, rats received control diet (Blue); CF, rats received control diet mixed with clofibrate (Yellow); HFD, rats received high-fat diet (Light green); HFD + CF, rats received high-fat diet mixed with clofibrate (Light red). Different letters indicate significant differences between groups (*p* < 0.05, significantly different by Tukey’s or Steel–Dwass multiple comparison test).

**Figure 6 biology-13-01027-f006:**
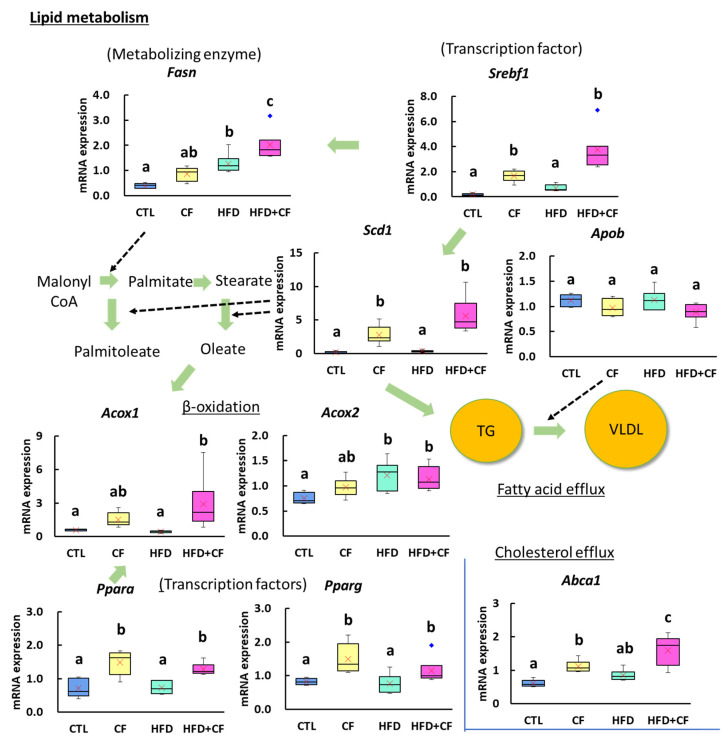
Gene expression analysis on lipid metabolism in the liver. *Fasn* and *Scd1* are genes regulating palmitate synthesis from malonyl CoA and palmitoleate and oleate synthesis from palmitate and stearate, respectively. *Srebf1* is a transcription factor gene for *Fasn* and *Scd1*. *Apob* is a gene of a product that plays a role in fatty acid efflux. *Acox1* and *Acox2* are genes regulating β-oxidation. *Ppara* and *Pparg* are transcription factor genes regulating β-oxidation and lipid metabolism. *Abca1* is a gene that plays a role in cholesterol efflux. Green arrows indicate lipid metabolic pathways, and dashed arrows indicate data associated with those pathways. Data represent a box-and-whisker plot (red cross, mean; blue dots, outliers). Group: CTL, rats received control diet (Blue); CF, rats received control diet mixed with clofibrate (Yellow); HFD, rats received high-fat diet (Light green); HFD + CF, rats received high-fat diet mixed with clofibrate (Light red). Different letters indicate significant differences between groups (*p* < 0.05, significantly different by Tukey’s or Steel–Dwass multiple comparison test).

**Figure 7 biology-13-01027-f007:**
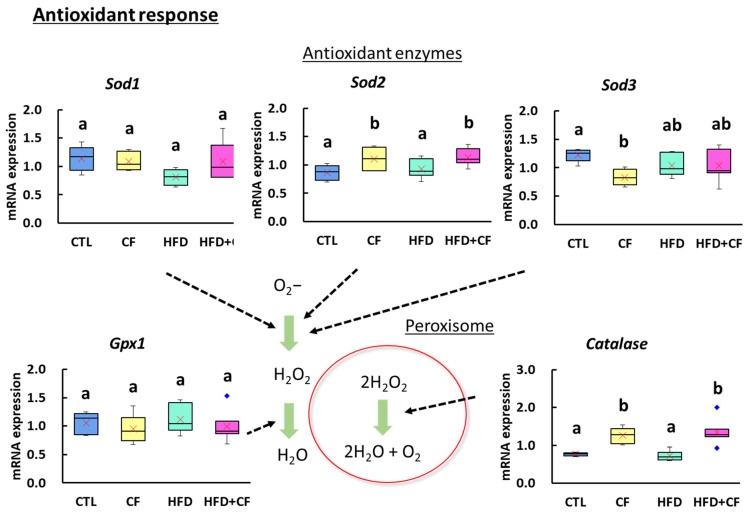
Gene expression analysis on antioxidant response in the liver. *Sod1*, *Sod2*, and *Sod3* are genes of enzymes synthesizing hydrogen peroxide from superoxide. *Gpx1* is a gene of an enzyme synthesizing hydrogen oxide from hydrogen peroxide. *Catalase* is a gene of an enzyme that synthesizes hydrogen oxide and oxygen from hydrogen peroxide. Green arrows indicate antioxidant responses and dashed arrows indicate data associated with those responses. Data represent a box-and-whisker plot (red cross, mean; blue dots, outliers). Group: CTL, rats received control diet (Blue); CF−, rats received control diet mixed with clofibrate (Yellow); HFD, rats received high-fat diet (Light green); HFD + CF, rats received high-fat diet mixed with clofibrate (Light red). Different letters indicate significant differences between groups (*p* < 0.05, significantly different by Tukey’s or Steel–Dwass multiple comparison test).

**Table 1 biology-13-01027-t001:** Final body weight, and, food and water intake in rats.

Group	CTL	CF	HFD	HFD + CF
No. of animals	5	6	6	7
Final body weight (g)	290.6 ± 10.5 ^a^	264.8 ± 11.4 ^b^	291.7 ± 6.5 ^a^	265.7 ± 9.5 ^b^
Food intake (g/rat/day)	12.1 ± 1.4 ^a^	12.2 ± 1.8 ^a^	8.5 ± 0.5 ^b^	9.7 ± 2.9 ^b^
Water intake (g/rat/day)	15.6 ± 2.7 ^a^	15.3 ± 2.8 ^a^	15.1 ± 3.7 ^a^	16.1 ± 2.9 ^a^
Abbreviations: BW, body weight; CF, clofibrate; CTL, control diet; HFD, high-fat diet.				
Data are shown as the mean ± standard deviation.				
Different letters indicate significant differences between groups (*p* < 0.05, significantly different by Tukey’s or Steel-Dwass test).				

**Table 2 biology-13-01027-t002:** Blood biochemistry in rats.

Group	CTL	CF	HFD	HFD + CF
No. of animals	5	6	6	7
AST (IU/L)	98.8 ± 8.8 ^ab^	151.2 ± 95.3 ^ab^	182.0 ± 83.5 ^b^	75.3 ± 7.9 ^a^
ALT (IU/L)	55.5 ± 3.7 ^a^	77.0 ± 60.0 ^a^	123.7 ± 57.2 ^a^	65.6 ± 13.5 ^a^
AST/ALT (ratio)	1.79 ± 0.22 ^ab^	2.07 ± 0.31 ^a^	1.49 ± 0.17 ^bc^	1.18 ± 0.18 ^c^
T-CHO (mg/dL)	56.3 ± 2.2 ^a^	55.8 ± 8.3 ^a^	61.8 ± 7.8 ^a^	55.1 ± 5.5 ^a^
F-CHO (mg/dL)	14.0 ± 0.8 ^a^	17.7 ± 2.6 ^b^	16.7 ± 1.0 ^ab^	18.1 ± 1.8 ^b^
E-CHO (mg/dL)	42.3 ± 1.7 ^a^	38.2 ± 6.7 ^a^	45.2 ± 7.2 ^a^	37.0 ± 4.6 ^a^
E/T (%)	75.0 ± 0.8 ^a^	68.2 ± 3.5 ^ab^	72.8 ± 3.4 ^ab^	67.0 ± 2.6 ^b^
TG (mg/dL)	60.0 ± 18.9 ^a^	32.5 ± 16.0 ^a^	41.0 ± 15.0 ^a^	31.9 ± 15.5 ^a^
NEFA (µEq/L)	644.0 ± 61.8 ^ab^	497.3 ± 106.4 ^a^	701.8 ± 176.6 ^b^	500.6 ± 111.4 ^a^
LDL-C (mg/dL)	4.5 ± 0.6 ^a^	5.5 ± 1.8 ^a^	4.7 ± 1.0 ^a^	6.3 ± 1.4 ^a^
HDL-C (mg/dL)	20.8 ± 1.0 ^ab^	20.5 ± 2.6 ^ab^	23.3 ± 2.4 ^b^	19.4 ± 1.8 ^a^
BUN (mg/dL)	20.7 ± 1.3 ^a^	26.1 ± 3.4 ^a^	24.8 ± 3.3 ^a^	26.9 ± 6.9 ^a^
CRE (mg/dL)	0.4 ± 0.1 ^a^	0.3 ± 0.0 ^ab^	0.4 ± 0.1 ^a^	0.3 ± 0.0 ^b^
Abbreviations: ALT, alanine aminotransferase; AST, aspartate aminotransferase; BUN, blood urea nitrogen; CF, clofibrate; CRE, creatinine; CTL, control diet; E-CHO, esterified cholesterol; F-CHO, free cholesterol; E/T, E-CHO/T-CHO; HDL-C, high-density lipoprotein cholesterol; HFD, high-fat diet; LDL-C, low-density lipoprotein cholesterol; NEFA, non-esterified fatty acid; T-CHO, total cholesterol; TG, triglyceride.				
Data are shown as the mean ± standard deviation.				
Different letters indicate significant differences between groups (*p* < 0.05, significantly different by Tukey’s or Steel-Dwass test).				

## Data Availability

The data generated or analyzed during this study are provided in this published article and Appendix A.

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
