# Peer review of "Peroxisome Proliferator Activator α Agonist Clofibrate Induces Pexophagy in Coconut Oil-Based High-Fat Diet-Fed Rats"

_biology, 2024, doi:10.3390/biology13121027_

Round 1
Reviewer 1 Report
Comments and Suggestions for Authors
-
This study examined the effects of the peroxisome proliferator-activated receptor-α agonist clofibrate on pexophagy in coconut oil-based high-fat diet (HFD)-fed rats. I have few comments prior the acceptance for publication.
- Abstract:
- To add more results (data and numbers) ref to data collected.
- Introduction:
- What is hypothesis of study?
- Methods:
- What are data of body mass of animals?
- Results:
- To add data regarding to body mass.
- Can authors to do correlation between Blood biochmistry and gene expression?
Discussion: well written
Author Response
Comment 1, Abstract: To add more results (data and numbers) ref to data collected.
(Answer) Thank you for your comments. The 200-character limit on the abstract makes it difficult to include test values, but we added FOLD CHANGES to the results, some of which are reflected in the ABSTRACT (L30-41)
Comment 2, Introduction: What is hypothesis of study?
(Answer) Thank you for your critical point. We revised the last sentence in INTRODUCTION to emphasize the hypothesis in this experiment (L106-123)
Comment 3, Methods: What are data of body mass of animals?
(Answer) Thank you for your critical point. Unfortunately, we could not calculate body mass index (BMI) in the present study. Please see the answer for comment 4.
Comment 4, Results: To add data regarding to body mass.
(Answer) Thank you for your critical point. Body mass index (BMI) is important to elucidate the growth and obesity of mice. BMI is calculated with body weight and body height (the length of nose to anus). In the present study, we did not measure the body height at necropsy. Evaluation of BMI in the model will be measured in future study.
References:
Gargiulo S, Get al., Biomed Res Int. 2014;2014:253067.
Comment 5, Can authors to do correlation between Blood biochmistry and gene expression?
(Answer) Thank you for your critical point. We compared selected parameters of blood biochemistry with gene expression levels of lipid metabolism and antioxidant responses. The data are shown in Figure S3-8 in Supplemental information. We confirmed that a high correlation was detected between T-CHO, F-CHO, E-CHO, E/T, LDL-C, and HGL-C and lipid metabolism (Abca1, Acox2, Apob, Ppara, and Pparg) and antioxidant response genes (Catalase, Gpx1, Sod1, Sod2, and Sod3) (L403-406; 482-485).
Reviewer 2 Report
Comments and Suggestions for Authors
Overall, this original research paper provides valuable insights into the effects of the peroxisome proliferator-activated receptor- α agonist clofibrate on pexophagy in coconut oil-based high-fat diet-fed rats. I have carefully reviewed the original research paper and have some constructive feedback for the authors:
The introduction is well-written, and the results support the discussion.
Major:
1. The methodology needs a more detailed description, especially regarding the procedures used. While a list of antibodies for immunohistochemistry and a sequence of primers used for RT-PCR are included in the supplement, a more comprehensive description of the procedures used is necessary.
2. Please clearly emphasize in the methodology (Animal Experiment section) the duration of clofibrate administration.
3. It would be helpful to know which anticoagulant was used for drawing the blood.
4. In the figure descriptions, please clearly explain the statistical significance of symbols 'a', 'b', or 'c'. For instance, specifying "a p < 0.05 significant difference vs. Control group" would add clarity.
Minor:
5. It is notable that only male mice were used in the study. Why were females not included?
Author Response
-
- The methodology needs a more detailed description, especially regarding the procedures used. While a list of antibodies for immunohistochemistry and a sequence of primers used for RT-PCR are included in the supplement, a more comprehensive description of the procedures used is necessary.
(Answer) Thank you for your critical point. We explained methods in detail on Animal Experiment, Histopathology, Ultrastructural Examination, Immunohistochemistry, and Real-Time Reverse Transcription-Polymerase Chain Reaction Analysis in supplemental methods.
- Please clearly emphasize in the methodology (Animal Experiment section) the duration of clofibrate administration.
(Answer) Thank you for your critical point. We revised the animal CF was administered during 2 and 14 weeks (for 12 weeks). We add Figure S1 to better understand the study design. We also explained the concept of the study in supplemental methods
- It would be helpful to know which anticoagulant was used for drawing the blood.
(Answer) Thank you for your critical point. We collected blood samples using heparinized syringes to isolate plasma (L152 and 153).
- In the figure descriptions, please clearly explain the statistical significance of symbols 'a', 'b', or 'c'. For instance, specifying "a p < 0.05 significant difference vs. Control group" would add clarity.
(Answer) Thank you for your critical point. We have explained in the last sentence in each fig legend as follows: Different letters indicate significant differences between groups (p<0.05, significantly different by Tukey’s or Steel-Dwass multiple comparison test). Adding a description of p value to each letter would be confusing, so we left the original proposal as is without modification.
Minor:
5. It is notable that only male mice were used in the study. Why were females not included? (Answer) Thank you for your critical point. We add the reason why we selected male rats in this study in L412-415 an Animal experiment in supplemental methods. We made a study plan on the base of a medium-term liver assay for 8 weeks to detect hepatocarcinogen for screening. The research groups examined strain, age, sex, initiation substance, timing of partial hepatectomy for enhancing preneoplastic lesions, and specific markers of preneoplastic foci to well and specifically detect hepatocarcinogenesis of test substances (Tamamo et al., 1983; Ito et al., 1997; 2003). According to these data, male F344rats were sensitive to the female as well as other strains (males). Importantly, in patients with NAFLD, males have a high risk of steatosis, NASH with expression of pro-inflammatory and pro-fibrotic cytokines, and HCC than females (Lonardo et al., 2019). That’s why we selected male rats in our HFD-mediated early hepatocarcinogenesis model.
References:
Ito, N., et al. Cancer Sci. 2003, 94, 3–8.
Ito N, et al. J Toxicol Pathol 1997; 10: 1-10.
Tamano S, et al. Cancer Lett. 1983 Oct;20(3):313-21.
Lonardo A, et al., Hepatology. 2019;70(4):1457-1469.
Reviewer 3 Report
Comments and Suggestions for Authors
Dear Editors and Dear Authors,
I read the article with great interest. In my opinion it is very valuable and the results deserve to be published.
Despite its great scientific value, I would like to suggest a few small corrections.
In the sentence:
"Cargoes, such as 52 damaged or aged intracellular organelles, are explicitly selected by receptor proteins, enabling selective macroautophagy [9] Pexophagy is a type of autophagy that selectively emoves excessive or damaged peroxisomes [10].",
punctuation mark, most likely a full stop after "[9]" is expected,
in contrast to sentence: "Kolmogorov-Smirnov test indicated that significant increases was detected between the CTL/CF/HFD. (p < 0.0083, modified with Bonferroni correction)" in which full stop in the middle in unnecessary.
I am not native speaker in english but some words as well as sentences seem to sound unusual e.g:
1) mentioned above word "emove" - is it really correct ?
2) "underwent liver initiation" - Could you explain what is the "liver initiation"?
3) "blood was collected from the abdominal aorta, and they were euthanized by blood release from the aorta and vein" - could you explain what kind of vein?
4) "29, 36,37" - I propose to standardize the system of spaces between the cited literature items.
5) Could you explain what is the "patial hepatectomy" ?
Nowadays, it is quite common to precisely state the p value in statistical tests rather than just stating that it is less or greater than 0.05.
It is worth providing information on the charts about what individual data means, e.g. whiskers etc.
Author Response
- "Cargoes, such as 52 damaged or aged intracellular organelles, are explicitly selected by receptor proteins, enabling selective macroautophagy [9] Pexophagy is a type of autophagy that selectively emoves excessive or damaged peroxisomes [10].",
punctuation mark, most likely a full stop after "[9]" is expected,
in contrast to sentence: "Kolmogorov-Smirnov test indicated that significant increases was detected between the CTL/CF/HFD. (p < 0.0083, modified with Bonferroni correction)" in which full stop in the middle in unnecessary.
I am not native speaker in english but some words as well as sentences seem to sound unusual e.g:
mentioned above word "emove" - is it really correct ?
(Answer) Thank you for your comments. In L62 and L345, we corrected the sentence and period according to the comments. We revised "emove" to "remove" in L62.
- "underwent liver initiation" - Could you explain what is the "liver initiation"?
(Answer) Thank you for your comments. We revised “liver initiation” to “initiation treatment” to avoid the confusion of treatment in L138. DEN is a genotoxic hepatocarcinogen, which induces DNA damage, providing initiating hepatocytes. We also revised the related sentence as “a two-step (initiation and promotion) hepatic carcinogenesis model” to understand the treatment of initiation and promotion in L136. The detailed concept of the study is also explained in supplemental methods.
- "blood was collected from the abdominal aorta, and they were euthanized by blood release from the aorta and vein" - could you explain what kind of vein?
(Answer) Thank you for your comments. We revised “vein” to “vena cava” in L153 and 154.
- "29, 36,37" - I propose to standardize the system of spaces between the cited literature items.
(Answer) Thank you for your comments. We corrected [29, 36,37] to [29,36,37] in L171.
- Could you explain what is the "patial hepatectomy" ?
(Answer) Thank you for your important comments. Partial hepatectomy is a surgical operation to increase cell proliferation activity of hepatocytes. This method also enhances the growth of preneoplastic lesions from DEN-initiated hepatocytes. The detailed concept of the study is also explained in supplemental methods.
Refferences:
Ito, N., et al. Cancer Sci. 2003, 94, 3–8.
Ito N, et al. J Toxicol Pathol 1997; 10: 1-10.
Tamano S, et al. Cancer Lett. 1983 Oct;20(3):313-21.
Lonardo A, et al., Hepatology. 2019;70(4):1457-1469.
- Nowadays, it is quite common to precisely state the p value in statistical tests rather than just stating that it is less or greater than 0.05.
It is worth providing information on the charts about what individual data means, e.g. whiskers etc.
(Answer) Thank you for your comments. We agree with your comments on statistical evaluation. In the present study, however, we evaluated four groups in each parameter, so it might be difficult to show all P values in all the data. In the revised figs, so we left the original proposal (p<0.05) as is without modification. According to another suggestion by the reviewer, we revised all Figs with a box-and-whisker diagram to better understand each data.
Reviewer 4 Report
Comments and Suggestions for Authors
In this manuscript titled “Peroxisome proliferator activator a agonist clofibrate induces pexophagy in coconut oil-based high-fat diet-fed rats”,the authors analyze the effects of clofibrate on hepatocytes and its molecular mechanisms.
This manuscript contains many interesting observations, but the purpose of each experiment is missing, and the results obtained are not presented in an easy-to-understand manner. Therefore, the results are not valid enough to support the authors' conclusions in this manuscript.
Comments:
1. Mate & Meth: The authors should describe the devices with which the sections were prepared and the details of the microscope (e.g., magnification used, name of manufacturer, etc.).
2. There is no scale bar in the Figure1 J.
3. The authors should write the purpose in each experiment. It should be clear what has been achieved and clarified. In particular, the reviewer is confused as to the purpose of measuring AST and ALT even though the addition of CF increased liver weight. Also, the purpose of analyzing the histology of the GFT-P is not clear.
4. In the lower left of Figure 1E, a vacuolar morphology is observed, is this typical of steatosis? The authors should describe the morphological characteristics in each condition.
5. In Figure 1 E, indicate by arrows the structures for the balloon-like changes and the areas where steatosis can be seen.
6. The authors did not observe autophagosomes clearly, so the "NBR1-positive autophagosomes" is incorrect. Please correct it.
7. In Figure 2, it is not clear where the antibody-positive granules are. The authors should clarify this by using arrows to indicate the location of the antibody-positive granules on the enlarged figures.
8. The illustration of the incorporation of peroxisomes within autophagosomes in Figure 4 is beyond the results. It is possible to mention it in the discussion, but the reviewer does not think it is appropriate in Figure 4.
9. In Figure 4-6, the term HF should be changed to HFD.
10. What do you think is degraded by autophagy (Figure 4), since the autophagy gene is activated?
11. Why does CF addition alone increase liver weight? Can this liver weight be considered aberrant? Could the slightly higher NAS and inflammation scores (F and I in Figure 1) or the higher expression of catalase (Figure 6) be related?
Author Response
- Mate & Meth: The authors should describe the devices with which the sections were prepared and the details of the microscope (e.g., magnification used, name of manufacturer, etc.).
(Answer) Thank you for your comments. We revised supplemental information to understand detailed histopathological and immunohistochemical examinations.
- There is no scale bar in the Figure1 J.
(Answer) Thank you for your comments. We revised Fig. 1J with bar.
- The authors should write the purpose in each experiment. It should be clear what has been achieved and clarified. In particular, the reviewer is confused as to the purpose of measuring AST and ALT even though the addition of CF increased liver weight. Also, the purpose of analyzing the histology of the GFT-P is not clear.
(Answer) Thank you for your critical comments. In RESULT, subtitles may be helpful to understand the aim and results. We also added the purpose in each experiment, sections 3.1-3.7.
- In the lower left of Figure 1E, a vacuolar morphology is observed, is this typical of steatosis? The authors should describe the morphological characteristics in each condition.
(Answer) Thank you for your comments. We revised Fig 1E at higher magnification to observe well steatosis. In L268-270, we explained fatty change in each group..
- In Figure 1 E, indicate by arrows the structures for the balloon-like changes and the areas where steatosis can be seen.
(Answer) Thank you for your comments. We revised Fig 1E at higher magnification to observe well balloon-like changes and steatosis (without arrows). We also add an inset of balloon-like changes in the HFD group.
- The authors did not observe autophagosomes clearly, so the "NBR1-positive autophagosomes" is incorrect. Please correct it.
(Answer) Thank you for your critical comments. We agree with your comments. In the ABSTRACT, we revised as “Pex5-, p62-, and LAMP2-positive granules and a decrease in NBR1-positive granules”.
We also revised the 3.4. subtitle in RESULTS as follows:
Coconut Oil-Based HFD Increases NBR1-Positive Granules, and HFD Combined with CF Administration Increased Pex5-, p62-, and LAMP2-Positive Granules.
.
- In Figure 2, it is not clear where the antibody-positive granules are. The authors should clarify this by using arrows to indicate the location of the antibody-positive granules on the enlarged figures.
(Answer) Thank you for your critical comments. Since there are numerous positive granules in images, an inset of the magnified image is included in each figure (the HFD+CF group in revised Fig. 2A, C, E, and G).
- The illustration of the incorporation of peroxisomes within autophagosomes in Figure 4 is beyond the results. It is possible to mention it in the discussion, but the reviewer does not think it is appropriate in Figure 4.
(Answer) Thank you for your critical comments. We agree with your comments, so we revised the illustration of not-incorporating peroxisomes within autophagosomes in revised Fig 5. To resolve the reviewer’s question, we confirmed peroxisomes captured by phagophore (isolated membranes) and autolysosomes, including peroxisomes in transmission electron microscopy (revised Figure 3, Figure S2).
- In Figure 4-6, the term HF should be changed to HFD.
(Answer) Thank you for your careful comments. We changed HF to HFD in revised Figure 5-7.
- What do you think is degraded by autophagy (Figure 4), since the autophagy gene is activated?
(Answer) Thank you for your critical comments. Our additional transmission electron microscopy experiment revealed that CF increased peroxisomes as well as pexophagy in the hepatocytes in CF and HFD+CF groups (revised Fig. 3 and Figure S2). We believe that hepatocytes enhance gene expression involved in autophagy (pexophagy) against numerous peroxisomes (revised Fig. 5), whereas at the same time, numerous peroxisomes are processed by pexophagy-regulated proteins (revised Figure 2). We revised the last sentence in DISCUSSION (L507-531).
- Why does CF addition alone increase liver weight? Can this liver weight be considered aberrant? Could the slightly higher NAS and inflammation scores (F and I in Figure 1) or the higher expression of catalase (Figure 6) be related?
(Answer)Thank you for your comments. Our additional electron microscopy experiment revealed that CF increased peroxisomes in the hepatocytes in CF and NFD+CF groups. Peroxisome proliferation increases liver weight. Catalase is one of the peroxisome enzymes, so the gene expression is closely related to peroxisome proliferation. We did not consider that slightly higher NAS and inflammation is related to increased liver weight.
References:
Berthier, A., et al., Biochim. Biophys. Acta. Mol. Basis Dis. 2021, 1867, 166097.
Meijer, J., et al., J. Ultrastruct. Mol. Struct. Res. 1989, 102, 87–94.
Miura, H., et al., Biophys. Acta. Mol. Cell Biol. Lipids 2021, 1866, 158963.
Round 2
Reviewer 2 Report
Comments and Suggestions for Authors
The methodology has been improved. Thank you.
Author Response
Thank you for your careful review. We appreciate your help in improving our manuscript.
Reviewer 4 Report
Comments and Suggestions for Authors
- What do you think is degraded by autophagy (Figure 4), since the autophagy gene is activated?
(Answer) Thank you for your critical comments. Our additional transmission electron microscopy experiment revealed that CF increased peroxisomes as well as pexophagy in the hepatocytes in CF and HFD+CF groups (revised Fig. 3 and Figure S2). We believe that hepatocytes enhance gene expression involved in autophagy (pexophagy) against numerous peroxisomes (revised Fig. 5), whereas at the same time, numerous peroxisomes are processed by pexophagy-regulated proteins (revised Figure 2). We revised the last sentence in DISCUSSION (L507-531).
(Reviewer) Page 9, line 307 – It is unclear what the increase in numbers is compared to what, and there is a lack of control or quantitative data. The authors should change the expression. It is also unclear whether the contents surrounded by autophagosomes are peroxisomes. The “Pex” organelles in Figure S2B appear to contain fractions that are not exclusively peroxisomes, such as ERs. The authors should change the “Pex” to “Phagophore” in the Figures 3 and S2.
Author Response
(Response)
Thank you for your thoughtful comments. We agree with the reviewer's comments.
In Page9, line 308, we revised the sentence as follows: Peroxisomes with dense cores and mitochondria were evident in the CF group (Figure 3A). Quantitative analysis of organelle in each group is essential, so we will make a plan to measure the numbers, size, and density of phagophore and autolysosomes in future studies.
We delete peroxisomes surrounded by sequestrated membranes” in Line 304-313. We revised Figures 2 and S2 to explain phagophores, as shown in revised Figures 3B and 3C and Figures S2B and S2D. We also revised the figure legends according to the these revised figures.
We also revised the sentence in Page 15, line 516 and 517 as follows: This was supported by the findings in TEM, in which organelles, possibly peroxisomes with phagophore expansion and autolysosomes were often observed.